# Data Science for Health Image Alignment: A User-Friendly Open-Source ImageJ/Fiji Plugin for Aligning Multimodality/Immunohistochemistry/Immunofluorescence 2D Microscopy Images

**DOI:** 10.3390/s24020451

**Published:** 2024-01-11

**Authors:** Filippo Piccinini, Marcella Tazzari, Maria Maddalena Tumedei, Mariachiara Stellato, Daniel Remondini, Enrico Giampieri, Giovanni Martinelli, Gastone Castellani, Antonella Carbonaro

**Affiliations:** 1IRCCS Istituto Romagnolo per lo Studio dei Tumori (IRST) “Dino Amadori”, 47014 Meldola, FC, Italy; marcella.tazzari@irst.emr.it (M.T.); maria.tumedei@irst.emr.it (M.M.T.); giovanni.martinelli@irst.emr.it (G.M.); 2Department of Medical and Surgical Sciences (DIMEC), University of Bologna, 40126 Bologna, BO, Italy; enrico.giampieri@unibo.it (E.G.); gastone.castellani@unibo.it (G.C.); 3Department of Physics and Astronomy “Augusto Righi” (DIFA), University of Bologna, 40127 Bologna, BO, Italy; m.stellato@unibo.it (M.S.); daniel.remondini@unibo.it (D.R.); 4Department of Computer Science and Engineering (DISI), University of Bologna, 47521 Cesena, FC, Italy; antonella.carbonaro@unibo.it

**Keywords:** histology/histopathology, immunohistochemistry/immunofluorescence techniques, multimodal micrographs, correlative microscopy, image registration

## Abstract

Most of the time, the deep analysis of a biological sample requires the acquisition of images at different time points, using different modalities and/or different stainings. This information gives morphological, functional, and physiological insights, but the acquired images must be aligned to be able to proceed with the co-localisation analysis. Practically speaking, according to Aristotle’s principle, “*The whole is greater than the sum of its parts*”, multi-modal image registration is a challenging task that involves fusing complementary signals. In the past few years, several methods for image registration have been described in the literature, but unfortunately, there is not one method that works for all applications. In addition, there is currently no user-friendly solution for aligning images that does not require any computer skills. In this work, DS4H Image Alignment (DS4H-IA), an open-source ImageJ/Fiji plugin for aligning multimodality, immunohistochemistry (IHC), and/or immunofluorescence (IF) 2D microscopy images, designed with the goal of being extremely easy to use, is described. All of the available solutions for aligning 2D microscopy images have also been revised. The *DS4H-IA* source code; standalone applications for *MAC*, *Linux*, and *Windows*; video tutorials; manual documentation; and sample datasets are publicly available.

## 1. Introduction

Biological samples are extremely complex systems that contain information at different scales. To gain a deep insight into pathologies and diseases, analyses at morphological, physiological, and functional levels are typically performed starting from histological slides [1]. For instance, in oncology, most tumour diagnoses go through some histology/histopathology analysis that is performed to describe and extract different features of the cancer cells which are illustrative of their in vivo behaviour [2]. Furthermore, in the era of immunotherapy, understanding the immune context beyond cancer cells is also pivotal for defining prognostic and predictive biomarkers [3]. This requires the acquisition of images with different modalities, different stainings, and most of the time, at different time points [4]. These give insights from different views, and to have a clear understanding of the patient’s situation, they must be merged into a single reference according to the old Aristotelian principle: “*The whole is greater than the sum of its parts*” [5]. Accordingly, image registration is an important task for medical diagnosis [6].

The term *image registration* refers to the general process of aligning image datasets. It became popular during the world wars with the need to align satellite and aircraft images [7]. In medicine and biology, alignment can involve overlapping images from different locations of the same sample or slide, or it can refer to images from different modalities (e.g., magnetic resonance and computed tomography), different time points (e.g., follow-up scans), and/or different subjects (in cases of population studies) [8]. However, spatial deformations may occur in the datasets. These can be caused by (*a*) microscope-induced aberrations, (*b*) required sample processing (e.g., drying) in between the image acquisitions, and (*c*) destructive techniques that consume or damage the sample. In such cases, the data will have to be acquired on consecutive histological sections, which naturally do not match perfectly due to the slightly different depths inside the respective specimen. This variety of factors that cause deformations and distortions in general requires flexible models for correction, and the need for correction is the reason behind the term *image co-registration* [9].

Today, the literature presents several methods for registering and co-registering images [10]. These can be case-specific, for instance, for registering images of two well-determined acquisition modalities, like PET and CT [11] or PET and MRI [12], or they can be generalist methods [1]. Specifically, for microscopy, the most common case of registration involving different acquisition modalities is known by the CLEM acronym, which stands for correlative (or correlated) light and electron microscopy [13]. This technique provides a unique possibility to correlate fluorescence microscopy data with ultrastructural electron microscopy information, thereby showing single molecules in the context of macromolecules, membranes, organelles, cells, and tissue. In this case, functional information and morphological information are correlated using a single reference to give a deep insight into the sample. A similar example or integration of microscopy morphological and functional data can be found when correlating optical light imaging techniques, such as brightfield, phase contrast, or DIC, with super-resolution fluorescence microscopy acquisitions. In these cases, most of the time, the images acquired are so different that just manual or semi-automatic registration methods can work [14]. Besides immunofluorescence (IF), image registration is also a fundamental step in immunohistochemistry (IHC), and it is the most common diagnostic technique used in oncology to analyse tissue pathology. IHC is characterised by the subsequent staining of a tissue section with markers requiring different sample preparations (sequential IHC or sIHC [15]) or the simultaneous detection of groups of multiple markers using stainings that can co-exist (multiplex IHC or mIHC [16]). The images obtained must then be registered using a single reference to gain a whole overview of the cancer tissue and be able to perform a co-localisation analysis (Figure 1), and most of the time, the registration method used should consider elastic deformations. To summarise, a multimodality registration method can be classified according to the following parameters: (*a*) modality-specific or generalist; (*b*) manual, semi-automatic, or automatic; and (*c*) rigid or elastic/deformable.

In this work, besides revising all of the solutions that are available for registering 2D microscopy images, the *Data Science for Health Image Alignment* (*DS4H-IA*) tool, a generalist open-source tool used for registering multimodality 2D microscopy images, is described. The tool is provided as a plugin for *ImageJ/Fiji* [17,18], one of the most common image analysis platforms for medical doctors, biologists, and life scientists in general. It was gradually created by gathering feedback from users with no advanced image processing experience [19], and thanks to the sample datasets, documentation, and instructional videos, the final release results were extremely user-friendly. Technically speaking, *DS4H-IA* provides strategies for registering images according to manual, semi-automatic, and fully automatic modalities, and it works with all of the medical imaging formats included in *Bio-Formats* [20]. In addition to being fully integrated with *ImageJ/Fiji*, it can also be used for performing elastic registration by passing the images to *bUnwarpJ* [21], a well-known ImageJ/Fiji plugin. The first command-line version of *DS4H-IA* was used by Bulgarelli et al. for aligning sIHC images [19]. Today, *DS4H-IA* has been extended with a user-friendly graphical user interface (GUI) and several opportunities to align the images [22]. The source code and compiled standalone versions for *Windows*, *Mac*, and *Linux* are freely provided at *www.filippopiccinini.it/DS4H-IA.html* (access date: 9 May 2023), together with sample datasets, documentation, and video tutorials.

The next sections are organised as follows: Section 2 presents a short overview of the methods available for registering 2D microscopy images. Section 3 describes the *DS4H-IA*. Section 4 presents the results obtained testing *DS4H-IA* with the representative datasets. Finally, Section 5 reports the main findings of the work.

## 2. Available Tools for Multimodality 2D Image Registration

Today, there are several tools used for registering multimodal 2D microscopy images. This section reports a brief description of their main features, and then summarises them in Table 1, Table 2 and Table 3. Figure 2 shows a screenshot of the different tools that have a GUI.

*Align image by line ROI*: *Align image by line ROI* (hereafter called *AIBLROI*) is a popular *ImageJ/Fiji* plugin created by Johannes Schindelin in 2006 using Java. *AIBLROI* is extremely easy to use; the user has to provide two landmarks per image by simply selecting a line. The order of the points is relevant: the first point will correlate with the first point of the other image’s line selection. *AIBLROI* has several limitations: (*a*) it handles only grey-level images; (*b*) it works with just two images at a time; and (*c*) it does not provide any output parameters to reproduce the registration result.

*BigWarp*: *BigWarp* is an *ImageJ/Fiji* plugin for manual, interactive, landmark-based deformable image alignment described in 2016 [23]. It uses point correspondences that are manually defined by the user through an interface that enables landmark pair placement and displays the effects of the warp on the fly. The registration model is then created using a thin plate spline to define the smoothest deformable transformation that exactly maps the landmarks, or according to the affine model (linear transform with translation, rotation, independent scales, and shear), the similarity model (linear transform with translation, rotation, and one scale parameter), the rotation model (also typically called rigid model, a linear transform with translation and rotation), or the simplest translation model (only displacements in *X* and *Y*). Landmarks can be exported and imported from plain text files. Once the warped image is obtained, it can be easily aligned with the reference image using the opportunities provided by *ImageJ/Fiji* (the authors provided several video tutorials to show how to effectively integrate *BigWarp* and *ImageJ/Fiji*).

*Correlia*: *Correlia* [9,24] is a platform-independent open-source *ImageJ/Fiji* plugin that is able to handle arbitrary 2D microscopy data and was specifically designed for the co-registration of 2D multimodal microscopy datasets. *Correlia* was developed at *ProVIS—Centre for Correlative Microscopy*. It was originally designed for the needs of chemical microscopy, involving various micrographs as well as chemical maps at different resolutions and fields of view. It comes with good documentation and sample datasets. On one hand, the software provides several manual and automatic registration opportunities, and it is also directly integrated with *bUnwarpJ* (another popular *ImageJ/Fiji* plugin) for performing elastic co-registration. On the other hand, it is not easy to use and often requires the definition of several parameters. It provides several visualisation opportunities but has limited exporting options. For instance, it requires the definition of a reference frame, which is then used to crop all of the images to be registered, and it does not allow for the aligned images to be exported as a full-resolution multi-frame stack.

*ec-CLEM*: *ec-CLEM* [25] is a free open-source software implemented as a plugin on the *Icy* platform [26]. The acronym *ec-CLEM* means “easy cell-correlative light to electron microscopy”, and it indicates the origin of the project: a tool for easily registering images acquired with light microscopes and electron microscopes. Today, the tool is able to work with a very wide variety of datasets, including 2D and 3D (or a mix of both dimensions) images. *ec-CLEM* offers several manual and automatic registration opportunities and works with time-lapse, multichannel, or multidimensional images. Registration can be carried out rigidly (only scale, rotation, and translation are applied in this case) or non-rigidly (nonlinear transformations based on spline interpolation, after an initial rigid transformation), and it automatically evaluates the need to apply non-rigid warping to obtain more accurate registration. Practically speaking, *ec-CLEM* is not easy to use and requires *Icy* to be installed on the computer, but it is a very interesting and wide solution for image co-registration. Video tutorials and extended documentation manuals are provided on the website.

*elastix*: *elastix* [8] is an open-source software based on *ITK* [27]. The software consists of a collection of algorithms that are commonly used to solve (medical) image registration problems. The modular design of *elastix* allows the user to configure, test, and compare different registration methods for a specific application. It has a modular design, including several optimisation methods, multiresolution schemes, interpolators, transformation models, and cost functions. The *C++* source code can be compiled on multiple operating systems (*Windows XP*, *Linux*, and Mac *OS X*). A command–line interface enables the automated processing of large numbers of datasets through scripting. Despite there being no official GUI, a few plugins exist for those who wish to use the functionality of *elastix* graphically. For instance, *SlicerElastix* (*https://github.com/lassoan/SlicerElastix*, access date: 9 May 2023) is an extension that makes *elastix* available in *3D Slicer* [28]. In addition, *elastix* is accompanied by *ITKElastix* (*https://github.com/InsightSoftwareConsortium/ITKElastix*, access date: 9 May 2023), making it available in *Python*, and by *SimpleElastix* (*https://simpleelastix.github.io/*, [29], access date: 9 May 2023), making it available in many languages like *Java, R, Ruby, C#*, and *Lua*. Finally, the authors also developed a version for *ImageJ/Fiji* (*https://imagej.net/plugins/elastix*, access date: 9 May 2023), but it is pretty difficult to install and difficult to successfully register the images.

*ITK*: The *National Library of Medicine Insight Toolkit* (*ITK*, [27]) is an open-source, cross-platform system used for medical image processing. It provides medical imaging researchers with an extensive suite of leading-edge algorithms for registering, segmenting, analysing, and quantifying medical data. It was conceived in 1999 to support the analysis of a specific project called *The Visible Human Project*. Then, it evolved into a technology underlying many medical image analysis commercial products worldwide. In 2005, the *ITK* community created a scientific journal, the *Insight Journal*, to fulfil the practice of the scientific method. In this journal, all articles are required to provide the full set of the source code, data, and parameters needed to reproduce the findings of the authors. Also, thanks to the *Insight Journal*, the *ITK* repository contains millions of lines of source code and is very popular today. However, it was designed for computer scientists/developers, and it is not a ready-to-use tool for biologists/medical doctors. In order to simplify the use of the *ITK*, the authors created *SimpleITK* [30], a simplified programming interface to the algorithms and data structures of the *ITK*. It supports interfaces for multiple programming languages, and not just the original *C++*. However, today, there is no official GUI. 

*Linear Stack Alignment with SIFT*: *Linear Stack Alignment with SIFT* (hereafter referred to as the acronym *LSAWSIFT*) is an *ImageJ/Fiji* plugin created by Stephan Saalfeld in 2008 after the publication of the scientific article by David Lowe introducing *SIFT* [31]. *LSAWSIFT* is a fully automatic registration algorithm based on a lightweight SIFT implementation for Java. It requires some settings and an input stack of images, and it provides, as the output, a new stack of images aligned with the first one, which is considered the reference one.

*Register Virtual Stack Slices*: *Register Virtual Stack Slices* (hereafter referred to as the acronym *RVSS*) is an *ImageJ/Fiji* plugin created by the same authors of *bUnwarpJ* [21]. *RVSS* was developed for co-registering a sequence of image slices stored in a folder by creating another list of registered image slices (with an enlarged canvas) according to one out of six pre-selected registration techniques: (*a*) Translation (only displacements in *X*, *Y*); (*b*) Rigid (translation and rotation); (*c*) Similarity (translation, rotation, and isotropic scaling); (*d*) Affine (translation, rotation, scaling, and shear); (*e*) Moving least squares (*https://imagej.net/plugins/moving-least-squares*, access date: 9 May 2023); and (*f*) Elastic (via *bUnwarpJ*). All models are aided by automatically extracted SIFT features. *RVSS* has a simple GUI but with advanced setup checkboxes for several registration options (the same as *LSAWSIFT*) and better integration with *bUnwarpJ*. In addition, the plugin also has the possibility to store the resulting transforms into “.xml” files, following the *TrakEM2* format [32]. This way, the results can be reproduced later on with the same images or in a different sequence. In addition, it is worth noting that the aligned images, provided as *RVSS*’output, are in full-size resolution without any loss in the data, image resize, or compression.

*StackReg*: *StackReg* [33] is an *ImageJ/Fiji* plugin for the recursive alignment of a stack of images. It was originally designed for registering a stack of slices from the same sample. Basically, images were acquired with the same imaging modality but referred to sections at different depths. Accordingly, *StackReg* is not optimised for registering multimodality images. In turn, each slice is used as the template to align the next slice, so that the alignment proceeds by propagation. The *StackReg* plugin requires a second plugin, named *TurboReg* (*http://bigwww.epfl.ch/thevenaz/turboreg/*, access date: 9 May 2023), to be installed. Then, five types of registration models are available, but none of them consider elastic transformations.

*TrakEM2*: *TrakEM2* [32] is an *ImageJ/Fiji* plugin for morphological data mining, three-dimensional modelling and image stitching, registration, editing, and annotation. In particular, for registration, it is designed for registering floating image tiles with each other using SIFT and global optimisation algorithms. Manual, semi-automatic, and fully automatic image registration are easily performed within and across sections according to one of the following registration models: (*a*) Translation; (*b*) Rigid; (*c*) Similarity; and (*d*) Affine. It also provides an algorithm for elastic alignment that compensates for nonlinear distortions [34]. In addition, the plugin also has the possibility to store the resulting transforms into “.xml” files and save and reload the projects. *TrakEM2* is a very wide tool, but it comes with detailed manuals and video tutorials to help all of the researchers who use it.

Excluding *DS4H-IA*, an analysis of Table 1 reveals that 8 out of 10 tools offer solutions for automatic alignment (all tools except *AIBLROI* and *BigWar*). Nearly all of them can be considered user-friendly, with the exceptions being *elastix* and *ITK* when used for automatic registration needs. It is noteworthy that *LSAWSIFT* lacks consideration for scale and rotation corrections, while *ec-CLEM* does not handle multiple images. Among the remaining tools, *Correlia*, *StackReg*, and *TrakEM2* permit manual corrections in the case of registration errors. However, it is important to note that none of these tools can output aligned images at their original size when dealing with high-resolution inputs. Consequently, it can be asserted that today, there is no freely available solution that is capable of consistently automatically registering multi-modal images while allowing for manual corrections and preserving the full size in the case of high-resolution images.

Finally, it is worth mentioning that today, the most popular commercial tools for image analysis (e.g., *IMARIS*, *ARIVIS*, *PATHCOREFLOW*, and *PHOTOSHOP*) and the most famous viewers used for histopathological image slides (e.g., *QuPath*, *Aperio ImageScope*, *Sedeen Viewer*, and *GIMP*) do not provide any opportunity for automatically aligning multimodal images.

**Table 3 sensors-24-00451-t003:** Tools for registering multimodality 2D microscopy images—scientific references.

*AIBLROI*	Not available.
*BigWarp*	Bogovic, J. A., Hanslovsky, P., Wong, A., & Saalfeld, S. (2016, April). Robust registration of calcium images by learned contrast synthesis. In 2016 IEEE 13th International Symposium on Biomedical Imaging (ISBI) (pp. 1123–1126). IEEE. [23]
*Correlia*	Rohde, F., BRAUMANN, U. D., & Schmidt, M. (2020). Correlia: an ImageJ plug-in to co-register and visualise multimodal correlative micrographs. Journal of Microscopy, 280(1), 3–11. [9]
*ec-CLEM*	Paul-Gilloteaux, P., Heiligenstein, X., Belle, M., Domart, M. C., Larijani, B., Collinson, L., … & Salamero, J. (2017). eC-CLEM: flexible multidimensional registration software for correlative microscopies. Nature methods, 14(2), 102–103. [25]
*elastix*	Klein, S., Staring, M., Murphy, K., Viergever, M. A., & Pluim, J. P. (2009). Elastix: a toolbox for intensity-based medical image registration. IEEE transactions on medical imaging, 29(1), 196–205. [8]
*ITK*	McCormick, M. M., Liu, X., Ibanez, L., Jomier, J., & Marion, C. (2014). ITK: enabling reproducible research and open science. Frontiers in neuroinformatics, 8, 13. [27]
*LSAWSIFT*	Not available.
*RVSS*	Not available.
*StackReg*	Thevenaz, P., Ruttimann, U. E., & Unser, M. (1998). A pyramid approach to subpixel registration based on intensity. IEEE transactions on image processing, 7(1), 27–41. [33]
*TrakEM2*	Cardona, A., Saalfeld, S., Schindelin, J., Arganda-Carreras, I., Preibisch, S., Longair, M., … & Douglas, R. J. (2012). TrakEM2 software for neural circuit reconstruction. PloS one, 7(6), e38011.
*DS4H-IA*	Piccinini, F., Duma, M.E. Tazzari, M., Pyun, J-C, Martinelli, G., Castellani, G., Carbonaro, A. (2022). DS4H Image Alignment: an user-friendly open-source ImageJ/Fiji plugin for aligning multimodality/IHC/IF 2D microscopy images. Submitted to Sensors.

## 3. Data Science for Health Image Alignment (DS4H-IA)

*DS4H-IA* is a generalist multimodal 2D image registration tool designed for medical doctors, biologists, and researchers in general with limited computer vision skills. It is an extremely user-friendly tool that is freely implemented in *Java* and provided as a plugin for *ImageJ* [17] and *Fiji* [18], which are the most common freely available image-processing platforms for quantitative microscopy analysis. Being integrated in *ImageJ/Fiji*, *DS4H-IA* exploits many other available plugins. For instance, it handles images of all the common medical imaging formats thanks to the *Bio-Formats* plugin [20]. In addition, it can also be used to perform elastic registration by passing its output images to *bUnwarpJ* [21]. The GUI of *DS4H-IA* version 1.0 is composed of multiple parts (Figure 2j): the current analysed image is visualised in the central window, and thanks to specific buttons, it is possible to scroll through the different uploaded images. The coordinates of the corners, which are manually defined in each image, are visualised on the top-left side, and thanks to some specific buttons, they can be deleted, modified, and copied into the other images. Finally, on the bottom-left side of the GUI, there are buttons used for registering the images according to manual, semi-automatic, and fully automatic modalities. All of the parameters can be set from the menu available on the bar at the top of the main GUI.

### 3.1. Registration—Via Corner Points

The images can be easily aligned by manually defining them with a few clicks and some well-visible reference marks (also called corner points) according to the classical translational, affine, and projective models [14]. At least one corresponding corner point for each image is required for the translative model, three corner points are required for the affine model, and four are required for the projective one. The tool then provides all of the facilities to easily move, modify, and copy the marks between the different images. The implemented least-squares [35] or random sample consensus algorithm (i.e., *RANSAC* [36]) automatically approximates the solution of the mathematically overdetermined system. The mathematical equations underlying the least-squares/*RANSAC* algorithm are reported in [37]. The coefficients estimated are then used to define the registration matrix for the alignment of the different images. The algorithm can also independently consider rotations and/or scale changes (to obtain rigid and similarity models), which is useful, for instance, in cases where the staining/destaining/stripping steps generate tissue dilation or shrinkage [38]. In cases of rotation or change in scale, the number of corresponding corner points increases to three for the translative models and remains at three and four for the affine and projective ones, respectively. Finally, thanks to an iterative subroutine for a fine alignment, the aligned images can be immediately loaded back to repeat the process and easily reach, in a few iterations, a very good image registration quality (Figure 3). Additional practical details on how to use the registration via corner points are reported in the user manual available at *www.filippopiccinini.it/DS4H-IA.html* (access date: 9 May 2023).

### 3.2. Registration—Automatic Modality

*DS4H-IA* provides an opportunity to completely automatically align 2D microscopy images. The designed approach is based on the well-known *SIFT* features (Scale-Invariant Feature Transform) [31] or, alternatively, on the *SURF* ones (Speeded Up Robust Features) [39]. Both the *SIFT*- and *SURF*-based algorithms work by detecting and describing key points in an image that are invariant to scale, rotation, and illumination changes. Briefly, starting from the open source code available in the *OpenCV* library [40], different solutions were implemented. They are selectable from the menus on the top part of the main GUI. The selected algorithm automatically computes the projection matrix to be used to align the subsequent images. Finally, a multichannel stack with all of the input images aligned in *z* is provided as the output. In addition, it can be reused as the input for *DS4H-IA* to (*a*) correct wrong alignments or (*b*) manually align images that are not easily handled by the automatic registration approach (e.g., DIC microscopy images when a dataset of fluorescence images is analysed). In these cases, it would be more precise to refer to a semi-automatic registration modality than a fully automatic one, because most of the images are automatically aligned using *SIFT/SURF*, but some of them are then registered via corner points that are manually defined by the user. An example of an output image (e.g., multichannel stack) automatically obtained using this approach is shown in Figure 4. Additional practical details on how to use the registration via automatic or semi-automatic modalities are reported in the user manual available at *www.filippopiccinini.it/DS4H-IA.html* (access date: 9 May 2023).

## 4. Experiments

*DS4H-IA* presents automatic, semi-automatic, and manual registration modalities. Thanks to these different opportunities, one of the main results is that with *DS4H-IA*, all of the partially overlapping images can be aligned. Practically speaking, the user can try to align the images automatically using the available *SIFT*- and *SURF*-based algorithms. In case these are not working properly, for instance, in case some images are not correctly automatically aligned, the user can (*a*) accept just the ones that are well aligned; (*b*) load back the images that are wrongly registered; (*c*) exploit the manual registration opportunity to define corner points that are visible in the overlapping images; and (*d*) align them with the previously automatically aligned images using the semi-automatic registration procedure. Otherwise, as the last but always working chance, the user can directly define corresponding corner points in all of the different images to be aligned and align them according to the manual registration procedure. Logically, aligning the images automatically significantly reduces the computation time required.

In order to validate *DS4H-IA*, a series of experiments using different multimodal datasets composed of partially overlapping images acquired in different modalities was performed. In particular, real-world images and synthetically generated ones have been used for testing *DS4H-IA*.

### 4.1. DS4H-IA Validation with Real-World Images

To start providing an idea of the reliability of the automatic registration modality, three different real-world datasets (i.e., *DatasetA*, *DatasetB*, and *DatasetC*) were used, each composed of five different series of four partially overlapping images acquired in different modalities (i.e., *Cy3*, *FITC*, *DIC*, and *Cy5*/*DAPI*). The five different series of each dataset were acquired manually, defining different positions of the microscope (i.e., *PositionA*, *PositionB*, *PositionC*, and *PositionD*), with a shift of approximately 25% of the images of the subsequent series, bringing an overlap of 75% between the images acquired in *PositionB* and *PositionA*, an overlap of 50% between the images acquired in *PositionC* and *PositionA*, an overlap of 25% between the images acquired in *PositionD* and *PositionA*, and moving back the microscope holder again to obtain an approximate overlap of 100% between the images acquired in *PositionE* and *PositionA*. For each position of the microscope, four images were acquired using different acquisition modalities: precisely, a fluorescent image in the wavelength of the *Cy3* channel, a fluorescent image in the wavelength of the *FITC* channel, a differential interference contrast (*DIC*) image, and a fluorescent image in the wavelength of the *Cy5* channel substituted just for *DatasetA* from one in the wavelength of the *DAPI* channel. All datasets refer to commercial histological samples sold by Salmoiraghi & Viganò (Botany #1, Legnago, Veneto, Italy), except for *DatasetA*, which is sold by Invitrogen (FluoCells #3, Waltham, Massachusetts, USA). The images were acquired using a Nikon A1R confocal microscope equipped with a 10× objective. *DatasetA* is composed of images referring to a mouse kidney section stained with *Alexa Fluor™ 488 WGA*, *Alexa Fluor™ 568 Phalloidin*, and *DAPI D-1306*; *DatasetB* is a whole-mount autofluorescence of silverberry scaly hair; and *DatasetC* is an autofluorescence transversal section of a leaf. Figure 5 shows the four different images acquired in *PositionA* and *PositionD* for the three different datasets. All of the experiments were performed using an entry-level Windows 64-bit PC (Intel Core i7 8th Gen, CPU 1.80 GHz, 12 GB RAM). The datasets are freely available for comparison with the results obtained with other tools or further analysis at *www.filippopiccinini.it/DS4H-IA.html* (access date: 9 May 2023).

Table 4 and Table 5 summarise the results obtained by visually evaluating the quality of the automatic alignment according to the *SIFT*- and *SURF*-based algorithms, respectively. When analysing the main diagonal, it is easy to appreciate that for the *SIFT*-based algorithm, in 83% of cases (i.e., 50 out of 60 cases), the registration of the images of the same acquisition modality successfully worked, even including *PositionD*, characterised by an approximative overlap of just 25% of the image. This value increases to 95% (i.e., 57 out of 60 cases) for the *SURF*-based algorithm. Considering the multimodal registration, for the *SIFT*-based algorithm, the *Cy3* image acquired in *PositionA* was correctly registered in 63% of cases (i.e., 38 out of 60 cases), the *FITC* image acquired in *PositionA* was correctly registered in 61% (i.e., 37 out of 60 cases), the *DIC* image was correctly registered in 18% (i.e., 11 out of 60 cases), and the *Cy5*/*DAPI* image was correctly registered in 38% (i.e., 23 out of 60 cases). By looking at Figure 5**,** it is possible to explain the poor values obtained for the *DIC* and *Cy5*/*DAPI* images: the *DIC* images are, in most cases, characterised by very low contrast, while the *Cy5*/*DAPI* images are, in most cases, complementary to the other fluorescent ones. Accordingly, it is particularly challenging to recognise the correct overlap between *Cy3*/*FITC* and these other signals. These results are also confirmed by the values achieved for the *SURF*-based algorithm, where the *Cy3* image acquired in *PositionA* was correctly registered in 56% of cases (i.e., 34 out of 60 cases), the *FITC* image acquired in *PositionA* was correctly registered in 56% (i.e., 34 out of 60 cases), the *DIC* image was correctly registered in 21% (i.e., 13 out of 60 cases), and the *Cy5*/*DAPI* image was correctly registered in 33% (i.e., 20 out of 60 cases). Despite the obvious difficulties when registering different, slightly overlapping images (e.g., images acquired in *PositionA* with images acquired in *PositionC* or *PositionD*), considering all cases together, the images were correctly automatically registered 45% of the time (i.e., 109 out of 240 cases) and 42% of the time (i.e., 101 out of 240 cases) for the *SIFT*- and *SURF*-based algorithms, respectively. Obviously, it is not possible to give general insights on registration performances without previously evaluating the content in the overlapping region of the image pair. Images that are mainly empty or flat are really challenging to align to failures in defining matching objects. However, roughly, without knowing the content of the images and the overlap, thanks to the experiments performed using real-world data, it is possible to claim that, on average, when using *SIFT*- and *SURF*-based algorithms (the most common ones among the tools proposed in the literature), the user should expect a correct automatic alignment for approximately 40% of the image pairs, making the semi-automatic procedure available in *DS4H-IA*, and causing it to be missing in most of the other tools, which is fundamental.

### 4.2. DS4H-IA Validation with Synthetically Generated Images

To objectively demonstrate the performance of *DS4H-IA*, two synthetically generated datasets of overlapping images were used. The first synthetic dataset (hereafter named *DatasetD*) was created by extracting overlapping images from the *DatasetA*—*PositionA Cy3* image, which is dense in content. The second synthetic dataset (hereafter named *DatasetE*) was created by extracting overlapping images from the *DatasetC*—*PositionA Cy3* image, an image that is practically black, so it is a really challenging case because registration algorithms typically fail in finding matches in empty regions. Each synthetically generated dataset is composed of three different images, overlapping to the reference one, respectively of 75%, 50%, and 25%. The root mean square error (RMSE, [14]) was computed in the overlapping portion that is common to all of the images to provide a quantitative measure of the registration accuracy. Figure 6 reports the four images of *DatasetD* and *DatasetE*, whilst Table 6 summarises the results obtained by aligning the reference image with the overlapping ones by using the *DS4H-IA SIFT*- and *SURF*-based algorithms for automatic registration. All synthetically generated images are available at *www.filippopiccinini.it/DS4H-IA.html* (access date: 9 May 2023), together with the MATLAB (The MathWorks, Inc., Natick, MA, USA) scripts used for creating the images and computing the RMSE values reported in Table 6. The obtained values show a great capability of *DS4H-IA SIFT*- and *SURF*-based algorithms to align the images, even in challenging cases. As expected, RMSEs related to images with an overlap of 25% are slightly higher than the ones related to 50% and 75%. The RMSEs related to *DatasetE*, lacking content, are better than the ones from *DatasetD* due to the presence of a black background, but they are characterised by a higher standard deviation (std). However, in the worst case (i.e., *DatasetD*, 25% overlap), the RMSE was lower than 1.

## 5. Conclusions

In many different application fields, ranging from surveillance to aerospace, there is a need to spatially align images acquired with different sensors or in different modalities. In particular, this is a very common problem in biology and medicine, where multimodality, immunohistochemistry, and immunofluorescence 2D microscopy images of the same sample are typically acquired to better understand pathological issues. However, the community is currently lacking a single method that is universally effective across all applications.

In this work, besides reviewing nine different freely available tools for aligning 2D multimodal images, *DS4H-IA*, an open-source, user-friendly tool specifically designed for aligning multimodal 2D images by exploiting different registration modalities ranging from fully automatic algorithms to manual ones, was described.

Despite *DS4H-IA* being a general-purpose tool, it was tested with different representative microscopy image datasets, proving that in approximately 40% of cases, it is able to automatically register multimodal images, even if those that are characterised by a very slight overlap. If, on one hand, this result is an appreciated output, on the other hand, it proves that a reliable manual modality is fundamental to obtain a solution for aligning all of the different images, and this is a missing opportunity in most of the tools that are available in the literature.

*DS4H-IA* is a modular and organised structure and open-source project developed using the Model–View–Controller (MVC) pattern, which strongly helps with extension. Accordingly, in the case of publishing new reliable automatic registration methods in the literature, it will be easy to include them in the tool (currently, *SIFT*- and *SURF*-based algorithms have been implemented). However, by offering several automatic, semi-automatic, and manual registration modalities, *DS4H-IA* currently represents the most complete open-source solution for those who need to align multimodality/IHC/IF 2D microscopy images for four main reasons: (*a*) it provides a solution to always be able to align every input image; (*b*) the aligned images are saved as full-resolution files without any compression or loss of information; (*c*) it presents an opportunity to perform elastic registrations, too; and (*d*) it is very easy to save and load back the project for further modifications or reanalyses.

*DS4H-IA* is implemented in *Java* and is distributed as an *ImageJ/Fiji* plugin. The *DS4H-IA* source code; standalone applications for *MAC*, *Linux*, and *Windows*; a video tutorial; manual documentation; and sample datasets are available at *www.filippopiccinini.it/DS4H-IA.html* (access date: 9 May 2023).

## Figures and Tables

**Figure 1 sensors-24-00451-f001:**
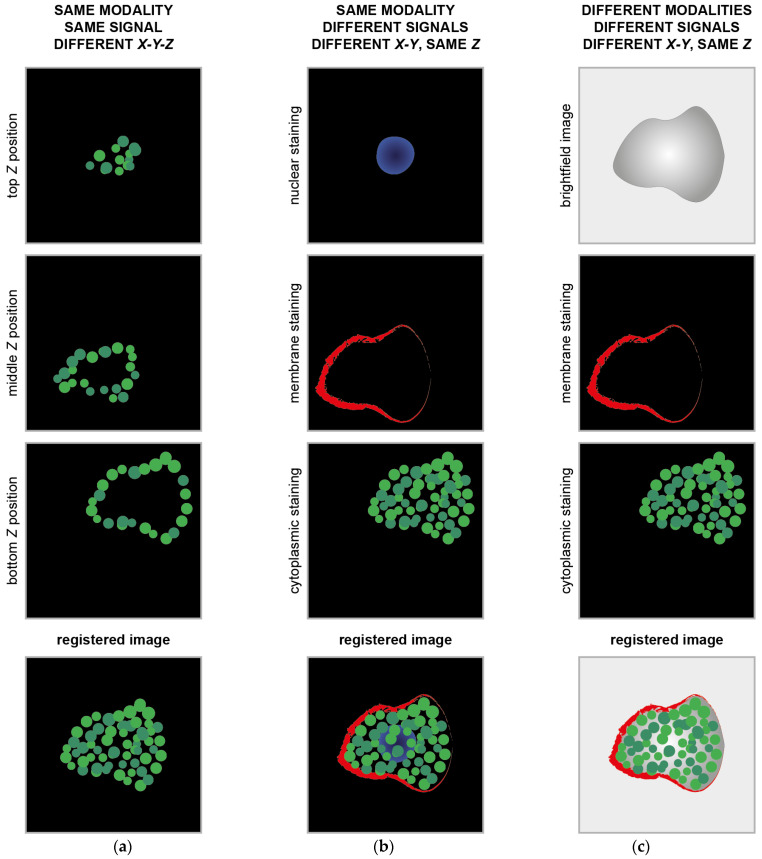
Example of classical image registration problems. (**a**) Images acquired at different Z planes. The cartoons mimic images of cytoplasmic proteins into a cell acquired using a fluorescence microscope with the centre of the images dislocated at different X-Y positions. (**b**) Images of different signals from the same object. The cartoons mimic images of subcellular compartments acquired using the same fluorescence microscope with the centre of the images dislocated at different X-Y positions. (**c**) Images of the same object acquired using different acquisition modalities. The cartoons mimic images of the same cell acquired with a brightfield microscope and a fluorescent microscope. The centre of the images is dislocated at different X-Y positions.

**Figure 2 sensors-24-00451-f002:**
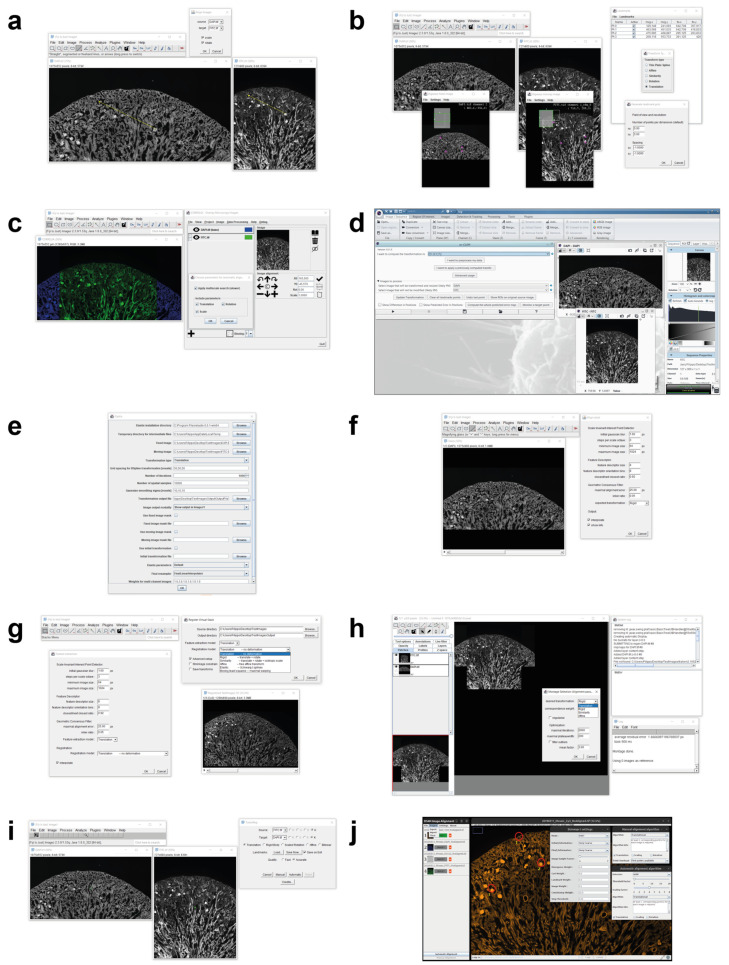
Main window screenshots of the different tools available today for registering multimodality 2D microscopy images. (**a**) *AIBLROI*, (**b**) *BigWarp*, (**c**) *Correlia*, (**d**) *ec-CLEM*, (**e**) *elastix*, (**f**) *LSAWSIFT*, (**g**) *RVSS*, (**h**) *TurboReg*, (**i**) *TrakEM2*, and (**j**) *DS4H-IA*.

**Figure 3 sensors-24-00451-f003:**
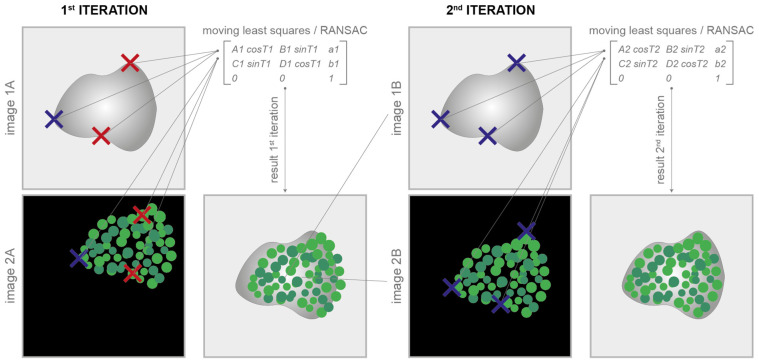
Registration via corner points. Corresponding corner points (represented in the cartoon as crosses) are manually defined in each image to be aligned. At the 1st iteration, it is not easy to perfectly define the correspondences (in the cartoon, wrong correspondences are represented with red crosses). The parameters of the transformation model are computed according to the least-squares/RANSAC algorithm. At the 1st iteration, the alignment result is, most of the time, unsatisfactory. Thanks to an iterative subroutine for a fine alignment, the aligned images can be immediately loaded back to repeat the process. In just a few iterations, it is pretty easy to reach a satisfactory result.

**Figure 4 sensors-24-00451-f004:**
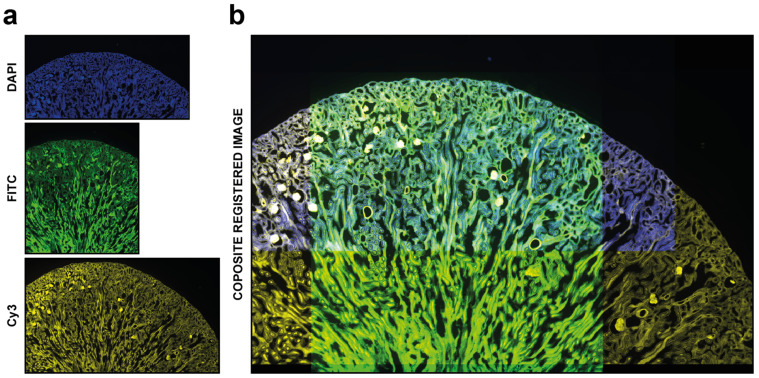
Automatic registration. Example of usage of *DS4H-IA*. (**a**) Input images referring to a commercial mouse kidney biopsy (FluoCell^TM^ prepared slide #3, Invitrogen). From top to bottom: Fluorescence *DAPI* (nuclear staining, *D-1306*), *FITC* (cytoplasmic staining, *Alexa Fluor 488* wheat germ agglutinin), and *Cy3* (membrane staining, *Alexa Fluor 568 phalloidin*) images all acquired using a Nikon A1R confocal Microscope equipped with a 20x objective. (**b**) Example of a common output registered stack. In this example, just the *DAPI*, *FITC*, and *Cy5* signals are shown.

**Figure 5 sensors-24-00451-f005:**
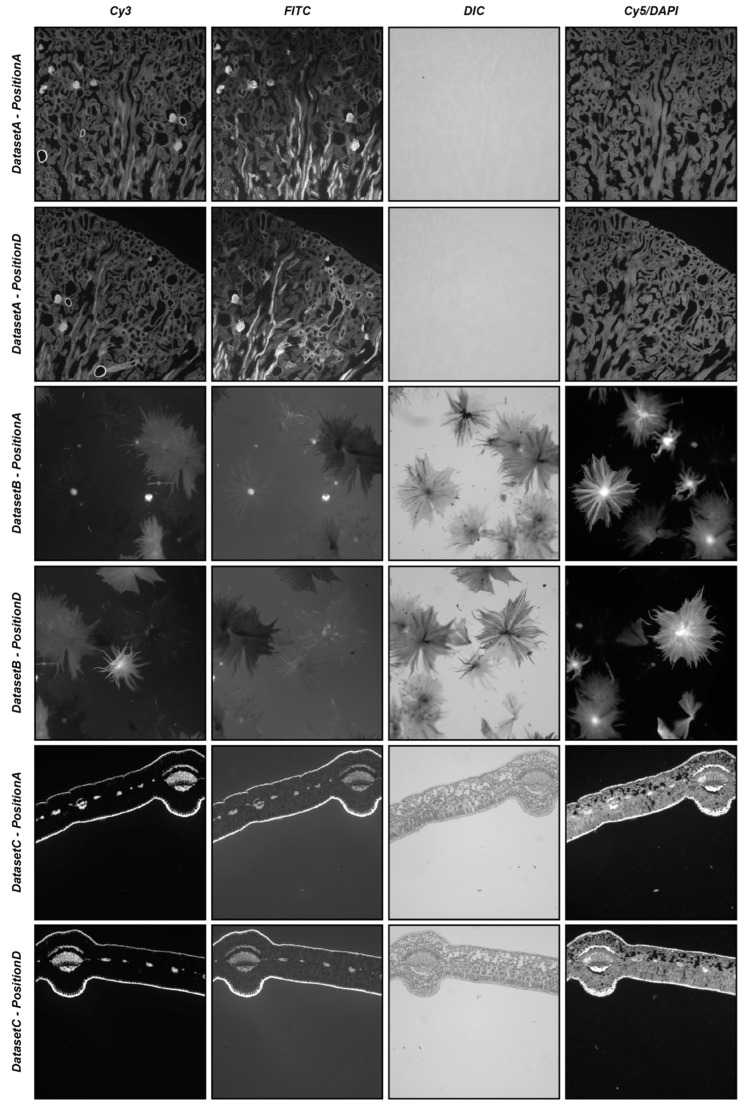
Example of images used in the experiments. From top to bottom: images from *DatasetA*, *PositionA*, and *PositionD*; *DatasetB*, *PositionA*, and *PositionD*; and *DatasetC*, *PositionA*, and *PositionD*. From left to right: images referring to the *Cy3*, *FITC*, *DIC*, and *Cy5/DAPI* signals.

**Figure 6 sensors-24-00451-f006:**
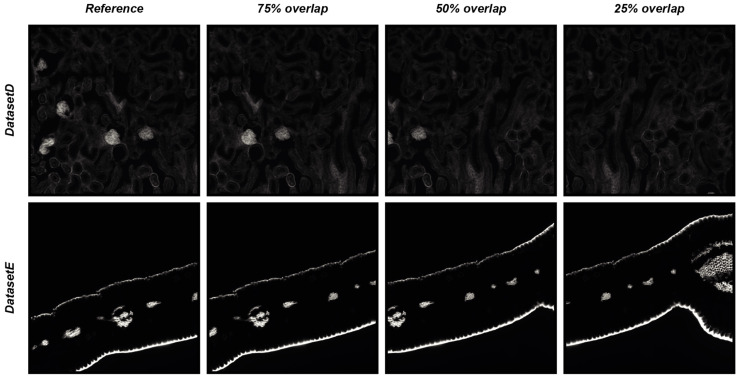
Synthetically generated datasets of overlapping images used for quantitative measure of registration accuracy of the *DS4H-IA SIFT*- and *SURF*-based algorithms for automatic registration.

**Table 1 sensors-24-00451-t001:** Tools for registering multimodality 2D microscopy images—characteristics (X = available/yes; O = not available/no)**.**

	*AIBLROI*	*BigWarp*	*Correlia*	*ec-CLEM*	*elastix*	*ITK*	*LSAWSIFT*	*RVSS*	*StackReg*	*TrakEM2*	*DS4H-IA*
VERSION											
Year of first release	2006	2016	2020	2017	2010	1999	2008	2009	2010	2005	2022
Current version	O	7.0.5	1.0	1.0.1.5	5.0.1	5.2.1	28 October 2018	3.0.7	7 July 2011	1.3.6	1.0
DOCUMENTATION											
User guide	X	X	X	X	X	X	O	X	X	X	X
Website	X	X	X	X	X	X	O	X	X	X	X
Video tutorial	O	X	O	X	X	X	O	O	O	X	X
Sample dataset	O	O	X	O	X	X	O	O	X	X	X
Open source	X	X	X	X	X	X	X	X	X	X	X
Implementation language	*Java*	*Java*	*Java*	*Java*	*C++*	*C++*	*Java*	*Java*	*Java*	*Java*	*Java*
USABILITY											
Input image format	All common	All common	All common	All common	All common	All common	All common	All common	All common	All common	All common
No programming experience required	X	X	X	X	O	O	X	X	X	X	X
User-friendly GUI	X	X	O	O	O	O	O	X	X	O	X
Intuitive visualisation settings	X	O	O	O	O	O	O	O	X	O	X
No commercial licences required	X	X	X	X	X	X	X	X	X	X	X
Portability on Win/Linux/Mac	X	X	X	X	X	X	X	X	X	X	X
FUNCTIONALITY											
Manual registration	X	X	X	X	O	X	O	O	X	X	X
Automatic registration	O	O	X	X	X	X	X	X	X	X	X
Image scale correction	X	X	X	X	X	X	O	X	X	X	X
Image rotation correction	X	X	X	X	X	X	O	X	X	X	X
Elastic correction	O	X	X	X	X	X	O	X	O	X	O
Multiple image handling	O	O	X	O	O	X	X	X	X	X	X
Multichannel/RGB image handling	O	X	X	X	X	X	X	X	X	X	X
OUTPUT											
Resized aligned images	X	X	X	X	X	X	X	X	X	X	X
Full-sized aligned images	O	O	O	O	O	O	O	X	O	O	X
Registration parameters	O	X	X	X	X	X	O	X	X	X	X
Editable result	O	X	X	X	X	X	O	X	O	X	X

**Table 2 sensors-24-00451-t002:** Tools for registering multimodality 2D microscopy images—download links (last accessed on 9 May 2023).

*AIBLROI*	https://imagej.net/plugins/align-image-by-line-roi
*BigWarp*	https://imagej.net/plugins/bigwarp
*Correlia*	https://www.ufz.de/index.php?en=47216
*ec-CLEM*	https://icy.bioimageanalysis.org/plugin/ec-CLEM/
*elastix*	https://en.wikipedia.org/wiki/Elastix_(image_registration)
*ITK*	https://itk.org/
*LSAWSIFT*	https://imagej.net/plugins/linear-stack-alignment-with-sift
*RVSS*	https://imagej.net/plugins/register-virtual-stack-slices
*StackReg*	http://bigwww.epfl.ch/thevenaz/stackreg/
*TrakEM2*	https://imagej.net/plugins/trakem2/
*DS4H-IA*	https://github.com/UniBoDS4H/DS4H-Image-Alignment

**Table 4 sensors-24-00451-t004:** Automatic alignment, *SIFT*-based algorithm, experimental result (X = well aligned; O = not aligned)**.**

		PositionA				PositionB				PositionC				PositionD				PositionE			
		Cy3	FITC	DIC	Cy5/DAPI	Cy3	FITC	DIC	Cy5/DAPI	Cy3	FITC	DIC	Cy5/DAPI	Cy3	FITC	DIC	Cy5/DAPI	Cy3	FITC	DIC	Cy5/DAPI
DatasetA - PositionA	Cy3	X	X	X	X	X	X	O	X	X	X	O	X	X	X	X	X	X	X	X	X
	FITC	X	X	X	X	X	X	O	X	X	X	O	X	X	X	X	O	X	X	X	X
	DIC	O	O	X	O	O	O	O	O	O	O	X	O	O	O	X	O	O	O	X	O
	DAPI	X	O	X	X	X	O	O	X	X	O	O	X	X	O	X	X	X	O	X	X
DatasetB - PositionA	Cy3	X	X	O	O	X	O	O	O	X	X	O	O	O	O	O	O	X	X	O	O
	FITC	X	X	O	O	O	X	O	O	O	O	O	X	X	O	O	O	X	X	O	O
	DIC	O	O	X	O	O	O	X	O	O	O	X	O	O	O	O	O	O	O	X	O
	Cy5	O	O	O	X	O	O	O	O	O	O	O	X	O	O	O	X	O	O	O	O
DatasetC - PositionA	Cy3	X	X	O	X	X	X	O	O	X	X	O	O	X	X	O	X	X	X	O	X
	FITC	X	X	O	O	X	X	O	X	X	X	O	X	X	X	O	X	X	X	O	O
	DIC	O	O	X	O	O	O	X	O	O	O	O	O	O	O	O	O	O	O	O	X
	Cy5	O	O	O	X	O	O	O	X	O	X	O	X	O	O	O	X	X	X	O	O

**Table 5 sensors-24-00451-t005:** Automatic alignment, *SURF*-based algorithm, experimental result (X = well aligned; O = not aligned)**.**

		PositionA				PositionB				PositionC				PositionD				PositionE			
		Cy3	FITC	DIC	Cy5/DAPI	Cy3	FITC	DIC	Cy5/DAPI	Cy3	FITC	DIC	Cy5/DAPI	Cy3	FITC	DIC	Cy5/DAPI	Cy3	FITC	DIC	Cy5/DAPI
DatasetA—PositionA	Cy3	X	X	O	X	X	X	O	X	X	X	O	X	X	X	O	X	X	X	O	X
	FITC	X	X	O	X	X	X	O	O	X	X	O	X	X	X	O	O	X	X	O	X
	DIC	O	O	X	O	O	O	O	O	O	O	O	O	O	O	O	O	O	O	O	O
	DAPI	X	O	O	X	X	O	O	X	X	O	O	X	X	O	O	X	X	O	O	X
DatasetB—PositionA	Cy3	X	X	O	O	X	O	X	O	X	O	O	O	X	O	X	O	X	X	O	O
	FITC	X	X	X	O	X	X	O	O	O	X	X	O	O	X	X	O	X	X	O	O
	DIC	O	X	X	O	O	O	X	O	O	O	X	O	O	O	X	O	O	O	X	O
	Cy5	O	O	O	X	O	O	O	X	O	O	O	X	O	O	O	X	O	O	O	X
DatasetC—PositionA	Cy3	X	X	O	O	X	X	O	O	X	X	O	O	X	X	O	O	X	X	O	O
	FITC	X	X	O	O	X	X	O	O	X	X	O	O	X	X	O	O	X	X	O	O
	DIC	O	X	X	O	O	O	X	O	O	O	X	O	O	O	X	O	O	O	X	O
	Cy5	O	O	O	X	O	O	O	X	O	O	O	X	O	O	O	X	O	O	O	X

**Table 6 sensors-24-00451-t006:** RMSE over the synthetically generated datasets of overlapping images.

	75% Overlap (Mean ± Std)		50% Overlap		25% Overlap	
	SIFT	SURF	SIFT	SURF	SIFT	SURF
*DatasetD*	0.01 ± 0.4	0.01 ± 0.4	0.02 ± 0.4	0.02 ± 0.4	0.03 ± 0.4	0.03 ± 0.4
*DatasetE*	0.02 ± 0.5	0.01 ± 0.5	0.03 ± 0.5	0.03 ± 0.5	0.04 ± 0.5	0.05 ± 0.5

## Data Availability

Part of the data presented in this study are available at: *www.filippopiccinini.it/DS4H-IA.html* (access date: 9 May 2023). All of the remaining ones are available upon request.

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
