# Peer review of "Data Science for Health Image Alignment: A User-Friendly Open-Source ImageJ/Fiji Plugin for Aligning Multimodality/Immunohistochemistry/Immunofluorescence 2D Microscopy Images"

_sensors, 2024, doi:10.3390/s24020451_

Round 1
Reviewer 1 Report
Comments and Suggestions for Authors
The manuscript titled "DS4H Image Alignment: a user-friendly open-source ImageJ/Fiji plugin for aligning multimodality/IHC/IF 2D microscopy images," submitted to 'Sensors', presents a valuable contribution to the field of image processing and microscopy. However, there are certain aspects that require major revision to enhance the quality and impact of the work. Below are three specific shortcomings identified in the manuscript:
1. The manuscript acknowledges the existence of several tools for multimodal 2D image registration but does not provide a detailed comparative analysis between DS4H Image Alignment (DS4H-IA) and these existing tools. Such a comparison is crucial to highlight the advantages and potential limitations of DS4H-IA. The authors should include a section that compares DS4H-IA with at least a few prominent existing tools in terms of usability, accuracy, and functionality. This comparison will not only underscore the unique features of DS4H-IA but also guide potential users in understanding the scenarios where this tool would be most beneficial.
2. The experiments section describes the use of DS4H-IA with different multimodal datasets, highlighting its capabilities in automatic, semi-automatic, and manual registration modalities. However, the manuscript lacks a comprehensive validation strategy. The authors should consider including quantitative measures of registration accuracy, such as root mean square error or other relevant metrics, to objectively demonstrate the performance of DS4H-IA. Additionally, a broader range of datasets, possibly including challenging cases where other tools fail or underperform, would significantly strengthen the validation process.
3. While DS4H-IA is described as a user-friendly tool designed for medical doctors, biologists, and researchers with limited computer vision skills, the manuscript does not adequately address the user experience and learning curve associated with the software. For a tool aimed at users without specialized knowledge in image processing, it is essential to provide data on ease of use, such as time taken for a novice user to become proficient, or feedback from real users who tested the tool in a research setting. Including such information would offer valuable insights into the practicality of DS4H-IA in a real-world scenario.
While the DS4H Image Alignment tool presents a promising solution for aligning multimodal 2D microscopy images, addressing these areas will significantly improve the manuscript. A thorough comparative analysis with existing tools, extensive validation with quantitative metrics, and detailed information on user experience will not only enhance the scientific rigor of the paper but also make it a more compelling and valuable resource for potential users in various fields of biological and medical research.
Comments on the Quality of English LanguageThe english language is ok.
Author Response
December 30, 2023
Bologna, Italy
Dear Editor,
We are submitting to Sensors a revised version of the manuscript with ID: “sensors-2744390”, and title: “DS4H Image Alignment: a user-friendly open-source ImageJ/Fiji plugin for aligning multimodality/IHC/IF 2D microscopy images”.
We sincerely thank the Reviewers for the suggestions that helped us to improve our work.
What follows is our detailed point-by-point reply to the comments received.
Looking forward to hearing from you in due course.
Thank you,
The corresponding author,
Filippo Piccinini
Filippo Piccinini, PhD
Senior Assistant Professor (RTD B), University of Bologna, Italy
Research Fellow, IRCCS IRST "Dino Amadori", Meldola, Italy
Email: f.piccinini@unibo.it
Mobile: +39 3495000398
REPLY TO REVIEWERS
Hereafter, “[C#]” stands for Reviewer’s comments and “[R#]” for our replies. All the reference numbers (pages, citations, etc.) refer to the revised version of the text, unless explicitly defined otherwise. In the answers below, the sign “< … >” refers to the text of the manuscript. In the manuscript, all changes are reported in red.
COMMENTS OF REVIEWER1
The manuscript titled "DS4H Image Alignment: a user-friendly open-source ImageJ/Fiji plugin for aligning multimodality/IHC/IF 2D microscopy images," submitted to 'Sensors', presents a valuable contribution to the field of image processing and microscopy. However, there are certain aspects that require major revision to enhance the quality and impact of the work. Below are three specific shortcomings identified in the manuscript:
[C1.1]
- The manuscript acknowledges the existence of several tools for multimodal 2D image registration but does not provide a detailed comparative analysis between DS4H Image Alignment (DS4H-IA) and these existing tools. Such a comparison is crucial to highlight the advantages and potential limitations of DS4H-IA. The authors should include a section that compares DS4H-IA with at least a few prominent existing tools in terms of usability, accuracy, and functionality. This comparison will not only underscore the unique features of DS4H-IA but also guide potential users in understanding the scenarios where this tool would be most beneficial.
[R1.1]
We thank Reviewer1 for the suggestion. The latest tool introduced in the literature for aligning multimodal 2D images is Correlia. Consequently, we chose to initiate a quantitative comparison between DS4H-IA and Correlia, utilizing synthetically generated datasets and employing the Root Mean Square Error (RMSE), as recommended by the same reviewer in [C1.2]. Unfortunately, Correlia demonstrated low qualitative correctness, aligning only one pair of images out of six. Anticipating potentially poorer results from older tools, we deemed it more appropriate to expand Section “2. Available tools for multimodality 2D image registration”, instead of adding a section on the quantitative analysis of competitor tools (which should have started by noting that Correlia was able to register only one dataset out of six, whereas DS4H-IA achieved alignment for them all).
[C1.2]
- The experiments section describes the use of DS4H-IA with different multimodal datasets, highlighting its capabilities in automatic, semi-automatic, and manual registration modalities. However, the manuscript lacks a comprehensive validation strategy. The authors should consider including quantitative measures of registration accuracy, such as root mean square error or other relevant metrics, to objectively demonstrate the performance of DS4H-IA. Additionally, a broader range of datasets, possibly including challenging cases where other tools fail or underperform, would significantly strengthen the validation process.
[R1.2]
Following the Reviewer's suggestion, we have created two synthetically generated datasets of overlapping images, one starting from an image dense in content, and one from an image practically black, basically a really challenging one where we expect tools to fail or underperform. The synthetically generated images are characterized by different overlapping percentages, precisely 75%, 50%, and 25%. To objectively demonstrate the performance of DS4H-IA, we have then used the root mean square error to provide quantitative measures of registration accuracy. All the details have been reported in the new section 4.2.
[C1.3]
- While DS4H-IA is described as a user-friendly tool designed for medical doctors, biologists, and researchers with limited computer vision skills, the manuscript does not adequately address the user experience and learning curve associated with the software. For a tool aimed at users without specialized knowledge in image processing, it is essential to provide data on ease of use, such as time taken for a novice user to become proficient, or feedback from real users who tested the tool in a research setting. Including such information would offer valuable insights into the practicality of DS4H-IA in a real-world scenario.
[R1.3]
We thank Reviewer1 for the suggestion that helped us to improve the manuscript. We modified the Introduction to better emphasize that:
(1) DS4H-IA was gradually created by gathering feedback from users with no advanced image processing experience;
(2) The sample datasets, documentation, and instructional videos, are freely provided to the community;
(3) DS4H-IA is designed as a plugin for ImageJ/Fiji, one of the most common image analysis platforms for medical doctors, biologists and life scientists in general.
The biologists involved in the project confirmed to us that immediately after reading the documentation and watching the video tutorials freely provided to become familiar with the different registration opportunities, it was extremely easy to work with DS4H-IA.
[C1.4]
While the DS4H Image Alignment tool presents a promising solution for aligning multimodal 2D microscopy images, addressing these areas will significantly improve the manuscript. A thorough comparative analysis with existing tools, extensive validation with quantitative metrics, and detailed information on user experience will not only enhance the scientific rigor of the paper but also make it a more compelling and valuable resource for potential users in various fields of biological and medical research.
[R1.3]
The Reviewer is right. In accordance with the recommendations, we conducted new studies utilizing quantitative metrics, enhanced our comparison with competitor tools, and made it more clear that the educational resources are publicly available.
[C1.5]
Comments on the Quality of English Language
The english language is ok.
[R1.5]
Thank you for all the suggestions.
COMMENTS OF REVIEWER2
[C2.1]
- Could you elaborate on the specific challenges or limitations encountered when attempting to achieve alignment for diverse or complex image sets?
[R2.1]
We thank Reviewer2 for the suggestion. To better elaborate on specific challenges or limitations encountered when attempting to achieve alignment for image datasets with a different complexity, we have created two synthetically generated datasets of overlapping images, one starting from an image dense in content, and one from an image practically black, basically a really challenging one. To objectively demonstrate the performance of DS4H-IA, we have then used the root mean square error to provide quantitative measures of registration accuracy. All the details have been reported in the new Section 4.2.
[C2.2]
- What strategies or considerations are in place to address the limitation that there isn't a single method universally effective across all applications?
[R2.2]
The comment of Reviewer2 made us realize that we should better emphasize that currently the community is lacking a single method universally effective across all applications and that DS4H-IA, offering several automatic, semi-automatic, and manual registration modalities, is the most complete solution today freely available for registering 2D images. The Conclusion Section has been modified accordingly.
[C2.3]
- In what specific scenarios or image characteristics does the tool heavily rely on manual intervention for successful alignment?
[R2.3]
Image pairs lacking content in the overlapping region are really challenging to be automatically aligned because the registration algorithms typically fail in finding matches. This has been now better emphasized in the new experiment performed, summarized in Section 4.2
[C2.4]
- Are there ongoing efforts to reduce dependency on manual intervention, especially in cases where automatic registration might be limited?
[R2.4]
Most likely, there is a miscommunication: our goal was not to reduce the dependency on manual intervention but to design DS4H-IA to make it easy.
[C2.5]
- Could you provide insights into the challenges faced in automatically registering images with minimal overlaps, as mentioned in the results (40% success rate)?
[R2.5]
The comment of Reviewer2 made us realize that we should better clarify that the 40% success rate is a rough estimation we obtained by analysing together image pairs with a different content and different overlaps. In fact, it is not possible to give general insights on registration performances without evaluating the content in the overlapping region of the image pair because logically Images mainly empty or flat are really challenging to be aligned to failures in defining matching objects. However, roughly, without knowing the content of the images and the overlap, thanks to the experiments performed using real-world data, we claimed that the user should expect a correct automatic alignment for approximately 40% of cases, making the semi-automatic procedure available in DS4H-IA fundamental.
[C2.6]
- What strategies or enhancements are being considered to improve alignment for images with very slight overlaps?
[R2.6]
Currently the literature is lacking reliable automatic registration methods for aligning images with very slight overlaps and the solution in these cases is requiring human intervention. Accordingly, we designed DS4H-IA to make manual intervention possible and easy to be performed.
[C2.7]
- What specific limitations or deficiencies were identified in the other available tools for aligning 2D multimodal images during the review process?
[R2.7]
The comment of Reviewer2 made us realize that while the main features of the different tools were summarized in Table 1, limitations and deficiencies were not enough emphasized. Accordingly, we extended the text of Section “2. Available tools for multimodality 2D image registration” with a new specific paragraph in this direction.
[C2.8]
- How does DS4H-IA address or aim to overcome these identified limitations of existing tools?
[R2.8]
Analysing Table 1 it is possible to see that 8 out of 10 tools provide solution for automatic alignment (i.e. all tools except AIBLROI and BigWar) and almost all of them can be considered as easy-to-use tools when automatic registration is needed (i.e. all except elastix and ITK). However, LSAWSIFT does not consider scale and rotation corrections and ec-CLEM does not handle multiple images. Among the remaining tools, just Correlia, StackReg, and TrakEM2 allow manual corrections in case of registration errors, but none of them is able to provide as output full-size aligned images in case of high-resolution inputs. Accordingly, we reported that today DS4H-IA is the sole freely available solution for being able to always automatically register multi-modal images by manually correcting for errors and keeping the full-size resolution. This has now been specified in Section “2. Available tools for multimodality 2D image registration”.
[C2.9]
- Could you elaborate on the planned improvements in implementing additional registration algorithms and incorporating new datasets?
[R2.9]
Including additional registration algorithms and incorporating new datasets is something not fundamental. However, DS4H-IA is a modular and organized structure open-source project developed using the Model-View-Controller (MVC) pattern, strongly helping for extension. Accordingly, in case of publication in the literature of new reliable automatic registration methods, it will be easy to include them in the tool. This has now been clarified in the text.
[C2.10]
- How do these enhancements intend to address current limitations or challenges faced by DS4H-IA in aligning multimodal images?
[R2.10]
To keep DS4H-IA updated by including future published reliable automatic registration methods is something nice. However, thanks to the semi-automatic procedure available in DS4H-IA, it is not fundamental.
[C2.11]
- How does DS4H-IA ensure the efficient handling of full-resolution images without compromising storage or processing resources?
[R2.11]
For high-resolution images, DS4H-IA automatically displays pop-up messages that guide users on extending the available memory from the ImageJ/Fiji menu up to the computer's maximum RAM capability.
[C2.12]
- Are there considerations in place to balance image quality preservation with resource efficiency?
[R2.12]
DS4H-IA does not compromise image resolution. However, users can logically utilize ImageJ/Fiji to resize the input images if needed.
[C2.13]
- What specific advancements or capabilities will the implementation of new automatic registration algorithms bring to DS4H-IA?
[R2.13]
To keep DS4H-IA updated by including future published reliable automatic registration methods is something nice. Nevertheless, due to the presence of a semi-automatic procedure within DS4H-IA, the incorporation of new automatic registration algorithms is not deemed indispensable.
[C2.14]
- How crucial is the diversity of datasets in assessing and improving the tool's performance across various imaging scenarios?
[R2.14]
The diversity of dataset in assessing and improving the tool's performance across various imaging scenarios is important. Logically, determining the optimal trade-off is challenging because on one hand using a wide range of datasets facilitates improvements in the tool's performance, but on the other hand complicates its assessment. In our study, we used six real-world image datasets and with two synthetically-generated ones. All the datasets have been publicly shared, fostering additional analysis and the evaluation of potential future tools.
[C2.15]
Comments on the Quality of English Language
Minor editing of English language required
[R2.15]
The English language has been improved. Thank you for all the suggestions.
COMMENTS OF SENSORS EDITORIAL OFFICE
[C3.1]
We noticed that a high proportion of the cited references belong to you or your co-authors: Refs. 8,9,17,18,26,44 and 45, which is a self-citation rate of about 16.67%. As MDPI is a member of COPE (https://publicationethics.org), all references in our published articles must contribute to the scholarly content of the paper and avoid bias (self-citations, journal citations, school of thought, etc.,) and reflect the current state of knowledge in the field. We encourage you to consider this and reduce the self-citations to make sure that only the most relevant citations are kept. If you feel that the citations to the previous works are essential, an Academic Editor will check the appropriateness of these citations prior to peer review.
[R3.1]
No problem: only references [18] and [26] are essential among these ones. Accordingly, references [8], [9], [17], [44], and [45] have been removed.

Reviewer 2 Report
Comments and Suggestions for Authors
- Could you elaborate on the specific challenges or limitations encountered when attempting to achieve alignment for diverse or complex image sets?
- What strategies or considerations are in place to address the limitation that there isn't a single method universally effective across all applications?
- In what specific scenarios or image characteristics does the tool heavily rely on manual intervention for successful alignment?
- Are there ongoing efforts to reduce dependency on manual intervention, especially in cases where automatic registration might be limited?
- Could you provide insights into the challenges faced in automatically registering images with minimal overlaps, as mentioned in the results (40% success rate)?
- What strategies or enhancements are being considered to improve alignment for images with very slight overlaps?
- What specific limitations or deficiencies were identified in the other available tools for aligning 2D multimodal images during the review process?
- How does DS4H-IA address or aim to overcome these identified limitations of existing tools?
- Could you elaborate on the planned improvements in implementing additional registration algorithms and incorporating new datasets?
- How do these enhancements intend to address current limitations or challenges faced by DS4H-IA in aligning multimodal images?
- How does DS4H-IA ensure the efficient handling of full-resolution images without compromising storage or processing resources?
- Are there considerations in place to balance image quality preservation with resource efficiency?
- What specific advancements or capabilities will the implementation of new automatic registration algorithms bring to DS4H-IA?
- How crucial is the diversity of datasets in assessing and improving the tool's performance across various imaging scenarios?
Minor editing of English language required
Author Response

(The authors gave the same response as above.)

Round 2
Reviewer 2 Report
Comments and Suggestions for Authors
no comment